# Endurance Exercise Does Not Exacerbate Cardiac Inflammation in BALB/c Mice Following mRNA COVID-19 Vaccination

**DOI:** 10.3390/vaccines12090966

**Published:** 2024-08-26

**Authors:** Sander Eens, Manon Van Hecke, Siel Van den Bogaert, Kasper Favere, Nathalie Cools, Erik Fransen, Tania Roskams, Hein Heidbuchel, Pieter-Jan Guns

**Affiliations:** 1Research Group Cardiovascular Diseases, GENCOR, University of Antwerp, 2610 Antwerp, Belgium; 2Laboratory of Physiopharmacology, GENCOR, University of Antwerp, 2610 Antwerp, Belgium; 3Laboratory of Translational Cell and Tissue Research, Department of Imaging and Pathology, University of Leuven, 3000 Leuven, Belgium; 4Department of Cardiology, Antwerp University Hospital, 2650 Antwerp, Belgium; 5Department of Internal Medicine, Ghent University, 9000 Ghent, Belgium; 6Laboratory of Experimental Hematology, Vaccine and Infectious Disease Institute, University of Antwerp, 2610 Antwerp, Belgium; 7Centre of Medical Genetics, University of Antwerp and Antwerp University Hospital, 2650 Antwerp, Belgium

**Keywords:** myocarditis, myopericarditis, mRNA vaccine, BNT162b2, endurance exercise, exercise, COVID-19, mouse, inflammation

## Abstract

The mechanism underlying myopericarditis associated with mRNA COVID-19 vaccination, including increased susceptibility in young males, remains poorly understood. This study aims to explore the hypothesis that engaging in physical exercise at the time of mRNA COVID-19 vaccination may promote a cardiac inflammatory response, leading to the development of myopericarditis. Male BALB/c mice underwent treadmill running or remained sedentary for five weeks. Subsequently, two doses of the Pfizer/BioNTech vaccine or vehicle were administered with a 14-day interval, while the exercise regimen continued. The animals were euthanized days after the second vaccination. Vaccination was followed by body weight loss, increased hepatic inflammation, and an antigen-specific T cell response. Small foci of fibrovascular inflammation and focal cell loss were observed in the right ventricle, irrespective of vaccination and/or exercise. Vaccination did not elevate cardiac troponin levels. Cardiac tissue from the vaccinated mice showed upregulated mRNA expression of the genes IFNγ and IL-1β, but not IL-6 or TNFα. This pro-inflammatory signature in the heart was not exacerbated by endurance exercise. Ex vivo vascular reactivity remained unaffected by vaccination. Our data provide evidence for the cardiac safety of mRNA COVID-19 vaccination. The role of exercise in the development of pro-inflammatory cardiac changes post mRNA vaccination could not be established.

## 1. Introduction

The coronavirus disease 2019 (COVID-19) pandemic precipitated a global health crisis with unprecedented socioeconomic disruption. In response, mass vaccination campaigns were rapidly implemented. Multiple COVID-19 vaccines emerged, including those developed by Pfizer-BioNTech (BNT162b2) and Moderna (mRNA-1273), utilizing first-of-their-kind messenger RNA (mRNA) technology. To date, billions of doses of these novel COVID-19 vaccines have been administered globally, with the general consensus supporting their efficacy and safety [1,2].

Shortly after the rollout of the COVID-19 vaccines, a number of rare but potentially life-threatening adverse events were reported, including Guillain-Barré syndrome, thrombosis–thrombocytopenia syndrome, and cerebral venous sinus thrombosis, among others [3]. Similarly, acute myocarditis and pericarditis associated with mRNA COVID-19 vaccination have garnered significant public attention [4]. Young males appear to be at highest risk, particularly after receiving two or more vaccine doses [5]. An (over)activation of the immune system and concomitant inflammatory cascade are suspected to be the underlying causes of these adverse events [6]. Despite extensive media coverage, the overall risk of myopericarditis after mRNA COVID-19 vaccination is considered low. A recent meta-analysis uncovered a combined incidence of 43.5 myopericarditis cases per million vaccine doses for both BNT162b2 and mRNA-1273 among adolescents aged 12–17 years [7]. The clinical presentation of these cases tends to be mild and self-limiting, although fatal instances with a fulminant phenotype have been reported [8].

So far, limited mechanistic insight into this novel association has been derived from a growing number of clinical case reports, and experimental research in this field is scarce. Interestingly, Li et al. reported that high-dose intravenous administration of the BNT162b2 mRNA vaccine consistently induced acute myopericarditis in BALB/c mice [9]. Nevertheless, the pathophysiology of this association remains poorly understood, including the factors that increase susceptibility in young male individuals. We hypothesize that physical exercise may be a potential factor that increases the risk of vaccine-related adverse inflammatory events, such as myopericarditis.

Physical exercise is recognized as a modulator of the immune system and inflammatory responses, and it has been suggested as acting as an adjuvant in the context of COVID-19 vaccination [10,11,12]. Although typically associated with beneficial effects, the immune-modulatory effects of physical exercise may lead to detrimental outcomes, such as in the context of viral myocarditis (i.e., increased myocardial inflammation and necrosis) [13,14,15,16,17,18]. Accordingly, current guidelines recommend complete exercise abstinence once myocarditis is suspected or diagnosed, regardless of its underlying cause [19]. The potential contributing role of physical exercise in cardiac inflammation or acute myopericarditis associated with mRNA COVID-19 vaccination, however, has yet to be investigated.

In the current study, we explored the hypothesis that exercising before or at the time of mRNA COVID-19 vaccination may induce or enhance a cardiac inflammatory response. To this end, we evaluated both the systemic and local immune responses to high-dose intravenous mRNA vaccination and their potential modulation by physical exercise.

## 2. Materials and Methods

### 2.1. Animals and Ethical Approval

A total of 96 five-week-old male BALB/c mice (OlaHsd substrain, Inotiv, The Netherlands) were studied. The animals were housed in standard cages (with a maximum of seven per cage) in a climate-controlled environment (20–24 °C, 45–65% relative humidity) under a 12 h light/dark cycle. Pelleted rodent chow and water were available ad libitum. All the tests were conducted during the light phase of the light/dark cycle. Before this study began, the animals underwent one week of acclimatization in the animal housing facility. This study’s protocol was approved by the University of Antwerp Ethical Committee for Animal Testing (approval nos. 2021-67 and 2022-68), adhering to the ARRIVE guidelines as well as the European Community Council Directive (2010/63/EU) on the protection of animals used for scientific purposes.

### 2.2. Study Design

At six weeks of age (D-7), the animals were randomly allocated to one of four groups (Figure 1A): vaccinated–sedentary (VAX-SED, *n* = 29), vaccinated–exercised (VAX-EEX, *n* = 29), or their respective vehicle controls (VEH-SED, *n* = 19; VEH-EEX, *n* = 19). The exercising groups underwent seven consecutive weeks of treadmill running (D0–D51), with exercise capacity evaluations conducted for all the animals at baseline (D-4) and during week 4 (D24). The mRNA COVID-19 vaccine or vehicle was administered in a two-dose regimen, with a 14-day interval between the doses (D35 and D49). Blood sampling and tissue collection were performed two to three days after the second injection of the vaccine or vehicle (D51–52).

### 2.3. Immunization

The animals were immunized intravenously through the lateral tail vein without anesthesia, utilizing a tubular-type mouse restrainer that provides free tail access. Each vaccine dose contained 6 µg of BNT162b2 mRNA (Pfizer/BioNTech, Puurs, Belgium; Comirnaty^®^; LOT: FH3219-01) and was diluted with normal saline to reach a total volume of 60 µL. The unvaccinated control groups received sham injections with an equal volume of normal saline. Our research group received a stock of BNT162b2 mRNA for experimental research from the federal government (FOD Volksgezondheid, Brussels, Belgium).

### 2.4. Treadmill Exercise Training and Maximal Running Speed

Forced treadmill running was conducted using motorized rodent treadmills (5 lane) equipped with an electric shock grid (LE8710MTS, Panlab, Barcelona, Spain). The exercising groups underwent treadmill running five days per week for seven consecutive weeks. In summary, each training session started with a 6 min warm-up, consisting of treadmill running for 3 min at 15 cm/s and 3 min at 20 cm/s. Next, the protocol continued for 54 min at the target speed of 30 cm/s, totaling 60 min of exercise training. To control for the possible non-exercise-mediated effects of treadmill running in the exercising groups, the sedentary groups were put on a slowly moving (5 cm/s) treadmill for 5 min during each training session. On the days of vaccination (D35 and D49), treadmill running was initiated within 2 h after injection. The treadmill inclination was consistently set at 0° for all the groups, and the continuous running of the animals was ensured through monitoring by a researcher. The mice who refused to run or who experienced frequent electric shocks in each training session (i.e., ≥250 shocks for three consecutive days), collectively referred to as “bad runners”, were excluded from this study. The exercise training was consistently conducted at the same time of day in a separate treadmill room, in the absence of other animal species.

A separate cohort of seven-week-old male BALB/cOlaHsd mice (*n* = 21) was utilized to determine the maximal running speed (V_max_). In brief, the V_max_ test began with treadmill running at 10 cm/s for 2 min, followed by increments of 10 cm/s every 2 min until the initial target speed of 50 cm/s was reached. The speed then continued to increase by 3 cm/s every 2 min until complete exhaustion, which was defined as residing for at least 5 s in the zone comprising the shock grid and one body length in front. The test was conducted twice, with two days of rest in between both trials, and the highest running speed achieved among both tests was defined as the V_max_.

### 2.5. Treadmill Exhaustion Test

Incremental exercise testing started at a speed of 12 cm/s and was further increased by 3 cm/s every 3 min until total exhaustion, which was defined as residing for at least 5 s in the zone comprising the shock grid and one body length in front. The treadmill inclination was set at 0°. Before baseline testing, the animals were familiarized with treadmill running for three consecutive days at a walking speed (5 min; 5 cm/s). Prior to the second exhaustion test, the exercising groups underwent 48 h of exercise training abstinence. For all the animals, both exhaustion tests were performed approximately at the same time of day in a separate treadmill room, in the absence of other animal species.

### 2.6. Euthanasia and Tissue Collection

The animals were anesthetized with a single intraperitoneal injection of sodium pentobarbital (150 mg/kg; Sanofi, Machelen, Belgium), followed by euthanasia through exsanguination (retro-orbital sinus). The blood was heparinized and centrifuged (2000× *g*, for 15 min, at 4 °C) to obtain the plasma. Following dissection, the hearts were submerged in ice-cold phosphate-buffered saline (PBS) with 30 mmol/L potassium chloride to remove any remaining blood and debris. The ventricular apex was snap-frozen in liquid nitrogen and stored at −80 °C for molecular analysis. The rest of the heart and liver were fixated in a 4% neutral-buffered formaldehyde solution for 24 h and then transferred into PBS until paraffin embedding. The whole spleens were collected and prepared as single-cell suspensions for flow cytometry, as will be further explained. In a subcohort of the animals, the thoracic aortas were dissected for an ex vivo evaluation of their vascular reactivity.

### 2.7. Organ Baths

The thoracic aortas were dissected into segments of 2 mm in length (i.e., TA0–TA5). The crossing of the diaphragm served as the reference point for the last segment (TA5). Segments TA2 or TA3 were mounted between two parallel wire hooks in a 10 mL organ bath set-up [20] containing a Krebs–Ringer solution with the following composition (in mmol/L): NaCl 118, NaHCO_3_ 25, glucose 11.1, KCl 4.7, CaCl_2_ 2.5, KH_2_PO_4_ 1.2, MgSO_4_ 1.2, and CaEDTA 0.025. The Krebs solution was kept at a constant temperature (37 °C) and equilibrated (95% O_2_/5% CO_2_) to maintain the desired pH (7.4). Isometric relaxations and contractions were measured by applying a fixed preload of 20 mN (approximating normal physiological stretch). In brief, cumulative concentrations of phenylephrine (PE) (3 nM–3 μM), an α_1_-adrenergic receptor agonist, were added to induce and assess vascular contractility. Additionally, endothelium-dependent relaxations were induced by a concentration–response stimulation of the segments with acetylcholine (Ach) (3 nM–3 μM), a muscarinic receptor agonist. The organ bath set-up allowed for eight segments to be run in parallel. Segments TA4 and TA5 were used for the molecular analyses.

### 2.8. Histology and Immunohistochemistry

Following (buffered) formalin fixation and paraffin embedding, the heart and liver tissues were cut into 5 µm sections and stained with hematoxylin and eosin (HE, 3801540BBE and 3801590BBE, Leica Biosystems, Wetzlar, Germany) for histopathological evaluation. The extent of cardiac inflammation was assessed by expert pathologists, using an in-house developed semiquantitative scoring system, considering the following categories: (a) no inflammation, (b) fibrovascular inflammation, and (c) fibrovascular inflammation accompanied by focal cell loss. Additionally, the location of the cardiac lesions was determined (left or right ventricle; subepicardial, midmyocardial, or subendocardial). For the liver, the sections were scored for the presence of portal and lobular inflammation. All the histopathological evaluations were performed in a blinded fashion.

For the immunohistochemical characterization of the cardiac infiltrates, a subset of the representative sections (*n* = 6) were stained with the following commercially available antibodies using a peroxidase technique: anti-F4/80 (diluted 1:100, ab6640, Abcam, Cambridge, UK), anti-iNOS (diluted 1:1000, ab15323, Abcam, Cambridge, UK), anti-Arginase1 (diluted 1:200, sc20150, Santa Cruz Biotechnology, Dallas TX, USA), and anti-CD3 (diluted 1:100, ab16669, Abcam, Cambridge, UK).

The expression of the SARS-CoV-2 spike protein in the hearts and livers was evaluated in a subset of the vaccinated animals (*n* = 6) using a rabbit polyclonal anti-SARS-CoV-2 spike protein antibody (diluted 1:500, NB100-56578, Bio-Techne Ltd., Minneapolis, MN, USA).

### 2.9. Cardiac Hypertrophy

Cardiac hypertrophy was assessed by the ‘mean nuclei count’ method using QuPath software v0.3.2 on digitalized HE slides (Zeiss Axio Scan.Z1, Oberkochten, Germany) [21]. Zones without evident inflammation or cell loss and with optimal cross-sectional cardiomyocyte orientation were identified in four left-ventricular regions (anterior, lateral, inferior, and septal). Next, the number of cell nuclei in a square of exactly 40,000 µm² were counted in each of the four zones, inversely correlated with cardiomyocyte volume. To validate this new semi-automated method, it was compared in a subcohort of the animals (*n* = 18 per group), with the established ‘mean linear intercept method’. In this method, the number of cardiomyocytes transected by a 250 µm line is counted manually in four different regions, as previously described [22]. Both methods showed a good correlation (Appendix A).

### 2.10. Plasma Cardiac Troponin I (cTnI)

The cTnI levels were determined for the plasma samples collected at euthanasia using the mouse CTNI ELISA kit (MBS766175, MyBioSource, San Diego, CA, USA), according to the manufacturer’s instructions. The target concentrations were interpolated from the standard curve using CurveExpert Professional software v2.7.3.

### 2.11. RNA Extraction

The total RNA was extracted from the heart and aorta (TA4-5) using the RNeasy^®^ Fibrous Tissue Mini kit (74704) and RNeasy^®^ Micro kit (74004, Qiagen GmbH, Hilden, Germany), respectively. The extractions were performed according to the manufacturer’s instructions. The RNA purity and concentration were assessed by a NanoDrop ND-2000 spectrophotometer (ThermoFisher, Waltham, MA, USA).

### 2.12. Reverse-Transcription Quantitative PCR (RT-qPCR)

The relative RNA expression was quantified by performing a two-step RT-qPCR using TaqMan™ Reverse Transcription Reagents (N8080234), TaqMan™ Universal PCR Master Mix (4304437), and Taqman™ primers (see below), according to the manufacturer’s instructions (ThermoFisher, Waltham MA, USA). Data normalization was performed using the following references genes: GAPDH (glyceraldehyde 3-phosphate dehydrogenase, Mm99999915_g1) and β-Actin (beta-actin, Mm00607939_s1). The following target genes were determined: ICAM-1 (intercellular adhesion molecule 1, Mm00516023_m), IFNγ (interferon gamma, Mm01168134_m1), IL-1β (interleukine-1 beta, Mm00434228_m1), IL-6 (interleukin 6, Mm00446190_m1), TNFα (tumor necrosis factor alpha, Mm00443258_m1), and VCAM-1 (vascular cell adhesion molecule 1, Mm01320970_m1). The plates were run on a QuantStudio™ 3 instrument (ThermoFisher, Waltham MA, USA). The relative expression levels were determined using the 2^(−∆∆CT)^ method.

### 2.13. Flow Cytometry and Intracellular Cytokine Staining

Upon dissection, the spleens were collected in a complete medium (RPMI 1640 medium + 10% FBS) and homogenized through a 70 µm cell strainer using the hard end of a 5 mL syringe plunger. After centrifugation, the splenocytes were treated for 5 min with red blood cell lysis buffer, re-centrifugated, and collected in FBS and 20% dimethyl sulfoxide (DMSO) for storage at −80 °C.

All the antibodies and reagents were from BioLegend Europe B.V. (Amsterdam, The Netherlands) unless otherwise specified. For the antigen-specific restimulation, approximately 1 × 10^6^ splenocytes were added to each well and stimulated for 6 h at 37 °C with 2.5 µg/mL SARS-CoV-2 peptide pools (S1 and S2) (PM-WCPV-S-1, JPT Peptide Technologies GmbH, Berlin, Germany), in the presence of 2 µg/mL anti-CD28 (102101) for costimulation. DMSO (1:200) served as the negative control, and a mixture of 50 µg/mL phorbol 12-myristate 13-acetate (PMA) and 1 µg/mL ionomycin served as the positive control. After 1 h of stimulation, a mixture of 5 µg/mL Brefeldin A (420601) and 2 µM Monensin (420701) was added for the inhibition of cytokine secretion. After 6 h of stimulation, the cells were washed with PBS, stained with LIVE/DEAD™ Fixable Near-IR (L34976, ThermoFisher Scientific, Waltham MA, USA), and Fc-blocked with TruStain FcX™ (anti-mouse CD16/32) antibody (156603). Following the Fc receptor blockade, the cells were surface marker stained for 20 min at 4 °C with the following antibodies: anti-CD3 AF700 (100215), anti-CD4 PerCP-Cy5.5 (100433), and anti-CD8 BV510 (100751). Next, the cells were washed with PBS + 0.1% BSA, fixed/permeabilized with a Cyto-Fast™ Fix/Perm buffer set (426803), and stained intracellularly for 30 min at 4 °C using the following antibodies: anti-IFN gamma APC (505809), anti-TNF alpha PE (506305), and anti-IL-4 BV711 (504133). The samples were run on a NovoCyte Quanteon flow cytometer and the data were analyzed using NovoExpress (v1.6.2) and FlowJo (v10.9.0) software.

### 2.14. Statistical Analysis

The normality of data was assessed both visually and by means of the Shapiro–Wilk test. The data are expressed as the mean ± SEM or as boxplots. The relevant statistical testing and significance levels are detailed in the figures or figure legends, as well as the sample sizes used for each statistical analysis. In summary, the changes in body weight between the groups over time and following vaccination were analyzed using a linear mixed model and a repeated measures two-way ANOVA, respectively. The group differences in the semiquantitative inflammation scoring for the liver and heart were assessed using logistic regression models. The experimental outcomes related to flow cytometry, cardiac troponin levels, gene expression, and vascular function were evaluated using the Kruskal–Wallis H test. The effects of exercise training on the groups were compared using two-tailed unpaired *t*-tests (* *p* < 0.05, ** *p* < 0.01, *** *p* < 0.001, and **** *p* < 0.0001). Notable near-significant *p* values (0.05 < *p* < 0.1) are additionally specified in the figures. The statistical analyses were computed using IBM SPSS Statistics v28.0 or GraphPad Prism v9.3.1. All the graphs were constructed using GraphPad Prism.

## 3. Results

### 3.1. BNT162b2 mRNA Vaccination Induced a Mild and Transient Decline in Body Weight

To explore the potential impact of exercise training on the development of cardiovascular injury following mRNA COVID-19 vaccination, groups of male BALB/c mice underwent continuous exercise training or remained sedentary and were administered two doses of BNT162b2 mRNA or a vehicle. The age-related body weight gain was similar among the experimental groups during the course of observation (Figure 1B). The initial injection of BNT162b2 mRNA induced a mild but significant decline in body weight in both the sedentary and exercising mice. Notably, this drop in body weight was transient, and recovered to the baseline level within four days (Figure 1C). Indications of pain, distress, or disease were absent across all groups throughout this study’s duration.

In total, 5 out of 96 mice were excluded from this study due to unexpected premature death (VEH-SED: *n* = 2, VEH-EEX: *n* = 1, VAX-SED: *n* = 2). Among these cases, four premature deaths were unrelated to the effects of vaccination and were likely attributable to the stress associated with either the restrained intravenous injections (VEH-SED, VEH-EEX) or the treadmill exhaustion testing (VEH-SED, VAX-SED). The fifth premature death (VAX-SED) occurred two days after the mouse received the second dose of the mRNA COVID-19 vaccine, with the necropsy revealing marked organomegaly. The immunohistochemical analyses of the affected organs ultimately led to a diagnosis compatible with B-cell lymphoblastic lymphoma, as reported separately [23]. However, it remains uncertain whether there was a causal link between the mRNA vaccine and the premature death, particularly since a decline in body weight had already been noted prior to vaccination.

### 3.2. Exercise Training Induced Physiologic Cardiac Remodeling

The exercising mice adhered to a fixed treadmill running protocol five days a week for seven consecutive weeks. The training sessions were conducted at a target speed of 30 cm/s, which corresponded to approximately 42% of their V_max_ (Appendix A). A functional assessment of exercise capacity by exhaustion testing during week four of exercise training did not reveal significant differences between the sedentary and exercising mice (Appendix A). Nevertheless, the sedentary mice showed a slight decrease in exercise capacity compared to their baseline measurements, while the exercising mice demonstrated a mild improvement. The completion of the full exercise training regimen induced physiologic cardiac remodeling, evidenced by increased heart-to-body weight ratios and decreased nuclei counts (indicative of larger cardiomyocyte sizes, i.e., cardiac hypertrophy) (Appendix A).

Three mice encountered difficulties in completing the exercise training regimen as required (i.e., ≥250 shocks for three consecutive days) during the initial five weeks of exercise training (i.e., before the first injection); consequently, they were excluded from this study.

### 3.3. BNT162b2 mRNA Vaccination Elicited Robust Antigen-Specific T Cell Responses

To assess the T cell responses elicited by BNT162b2 mRNA vaccination, splenocytes from all four groups were isolated and ex vivo stimulated with peptide pools overlapping each subunit of the spike protein (S1 and S2), followed by intracellular cytokine staining (Figure 2A). The flow cytometry gating strategy is illustrated in Figure 2B. Vaccination induced high frequencies of S1-specific CD8^+^ T cells expressing the Th1 cytokines IFNγ and TNFα (Figure 2C,D). Conversely, within the CD4^+^ T cell population, this Th1-polarized response was observed upon stimulation with both peptide pools. Exercise training did not modulate these antigen-specific CD4^+^ or CD8^+^ T cell responses. No discernible increases in S-specific T cells expressing the Th2-type cytokine IL-4 could be detected following vaccination (Figure 2E).

### 3.4. BNT162b2 mRNA Vaccination Increased the Occurrence of Minimal Hepatic Inflammation

At the necropsy, no macroscopic aberrations were observed in the livers. The histopathological examination revealed very small foci of portal and lobular inflammation across all the groups, with lesions more frequently observed in the mice administered the BNT162b2 mRNA (Figure 3A,B). Exercise training did not influence the prevalence of these lesions, and no significant differences in density or composition were noted between the lesions in the vaccinated and unvaccinated mice. Liver expression of the SARS-CoV-2 spike glycoprotein was not detected; however, the same antibody did show positivity in the lung tissue of a SCID mouse that experienced a confirmed SARS-CoV-2 infection.

### 3.5. BNT162b2 mRNA Vaccination, Alone or Combined with Exercise Training, Did Not Induce Cardiac Inflammation or Cell Loss upon Histopathological Examination

At the necropsy, no gross cardiac abnormalities were observed. The histopathological evaluation of the HE-stained heart sections revealed discrete foci of fibrovascular inflammation (7.4–23.1%) and fibrovascular inflammation accompanied by focal cell loss (3.7–11.5%) across all the groups, irrespective of their vaccination or exercise status (Figure 4A,B). These lesions were exclusively located subepicardially in the right ventricle, except for one animal in the VAX-EEX group that showed a left-ventricular injury.

An immunohistochemical characterization of the inflammatory foci revealed a similar density and composition of the infiltrates across all the groups, predominantly consisting of macrophages (F4/80^+^) of the pro-inflammatory subtype (iNOS^+^). The sporadic presence of lymphocytes (CD3^+^) was observed (Figure 4C). Intracardiac expression of the SARS-CoV-2 spike protein was not detected.

Myocardial injury was further assessed by evaluating the cTnI plasma levels (Figure 4D). Consistent with the histopathological observations, no notable differences in plasma cTnI concentrations were observed across the groups. Three animals were excluded from this analysis due to unsuccessful blood sampling.

### 3.6. BNT162b2 mRNA Vaccination Upregulated Cardiac Pro-Inflammatory Cytokine Expression, Regardless of Exercise Status

Vaccination induced a significant cardiac upregulation of IFNγ (sixfold) and IL-1β (threefold) gene expression (Figure 4E,F). Moreover, the expression levels of these genes tended to be higher in the sedentary vaccinated mice compared to their exercising counterparts, although these differences were not statistically significant. No clear signal for the differential expression of IL-6 was observed (Figure 4G), whereas TNFα showed elevated expression levels in both vaccinated groups compared to the VEH-EEX (Figure 4H). Exercise alone did not significantly alter gene expression across all the pro-inflammatory cytokines.

### 3.7. BNT162b2 mRNA Vaccination Caused Subtle Alterations in Vascular Reactivity

The ex vivo vascular reactivity of the aortic segments from the vaccinated and unvaccinated mice was assessed through concentration–response stimulation with PE and ACh, respectively. No changes in PE-induced vascular contractility were observed across the groups (Figure 5A). Both vaccinated groups showed a tendency toward enhanced endothelium-dependent relaxation and an increased sensitivity to ACh compared to their unvaccinated counterparts, although these trends were not statistically significant (Figure 5B). Exercise did not have a modulating effect on vascular reactivity. Figure 5C,D illustrate the effect of exercise and vaccination on the expression of endothelial adhesion molecules ICAM-1 and VCAM-1 in the aortic tissue. Visually, there appears to be a tendency toward the upregulation of these genes following vaccination (for ICAM-1 and VCAM-1) and exercise (for ICAM-1 only), but this is not statistically significant.

## 4. Discussion

Myopericarditis has emerged as a rare, yet potentially severe complication associated with mRNA COVID-19 vaccination, predominantly affecting adolescent and young adult males [24]. To date, the mechanisms underlying this novel association remain poorly understood. In the present study, we hypothesized that engaging in physical exercise before or at the time of mRNA COVID-19 vaccination might promote an inflammatory response in the heart, potentially serving as a substrate for the development of acute vaccine-related myopericarditis. However, our findings do not support this hypothesis, as treadmill running in mice did not augment the mild cardiac inflammation elicited by mRNA COVID-19 vaccination. Interestingly, exercise training rather tended to decrease the expression of cardiac pro-inflammatory cytokines after vaccination, albeit not to a statistically significant degree.

Effective vaccination was evidenced by the transient body weight loss; increased hepatic inflammation; and a robust, spike-specific, T cell response. The transient decline in body weight following mRNA vaccination is likely due to a general malaise induced by systemic inflammation, leading to reduced food intake, particularly given the intravenous administration route and the markedly high dosing. Biodistribution studies have demonstrated a substantial accumulation of intramuscularly injected nucleoside-modified mRNA in the liver [25,26]. This accumulation likely results from the systemic distribution of lipid nanoparticles, and could be significantly amplified with intravenous administration. While mRNA COVID-19 vaccine-induced liver injury has been documented, particularly in autoimmune contexts, the underlying pathophysiologic mechanisms remain poorly understood [27,28]. Additionally, it is important to note that the inflammatory foci observed in our study were discrete, and vaccination did not affect the density or composition of the infiltrates.

The strong CD8^+^ T cell response and IFNγ^+^ signature align with previous observations following administration of the BNT162b2 mRNA vaccine [29,30]. Consistent with the findings of Corbet et al. [31], the CD8^+^ T cell responses were exclusively directed against the S1 subunit of the SARS-CoV-2 spike protein, while the CD4^+^ T cells exhibited mild reactivity to both peptide pools. Notably, the chronic exercise training regimen did not modulate the systemic T cell response elicited by mRNA COVID-19 vaccination, which challenges the concept of physical exercise as an adjuvant that enhances the vaccination response [10]. However, we did not quantify the antibody titers of the mice, and need to acknowledge the limitation that our study employed only one type of exercise training. It is likely important to take variations in training characteristics (i.e., duration, intensity, and timing) into consideration before reaching any final conclusions.

Overall, high-dose intravenous mRNA COVID-19 vaccination did not induce myopericarditis, which contrasts with the findings of a previous report [9]. Regardless of vaccination and exercise status, our histopathological examination uncovered minimal cardiac fibrovascular inflammation and localized cell loss in a subset of animals. These discrete, iNOS macrophage-dominant lesions typically presented as solitary inflammatory foci, and were nearly exclusively located within the right-ventricular epicardium. The histopathological features of these lesions may raise suspicions of an end-stage manifestation of spontaneous cardiac calcinosis, a condition to which the BALB/c strain is particularly susceptible [32,33,34]. Consistently, plasma cTnI levels did not increase with vaccination and/or exercise.

Although the histopathology did not reveal changes consistent with myopericarditis, the RT-qPCR analysis of the pro-inflammatory genes demonstrated a clear vaccination response in the heart. Specifically, cardiac tissue from the vaccinated mice exhibited a prominent upregulation of IFNγ and IL-1β gene expression. These cytokines are well-established players in the immune response elicited by both SARS-CoV-2 infection and vaccination [35,36,37]. Given that cardiomyocytes generally do not produce IFNγ, the upregulation of this gene likely results from the activation of cardiac tissue-resident cells and/or infiltrating immune cells (in the context of a systemic inflammatory response to vaccination) [38]. Notably, the expression levels of TNFα and IL-6 remained unaltered, despite the typical crosstalk among the studied cytokines [39].

Interestingly, vaccination triggered subtle changes in the aorta regardless of exercise status. The observed tendency toward the upregulation of VCAM-1 suggests mild endothelial activation [40], although statistically significant differences were not demonstrated. Signs of vascular dysfunction were absent. Conversely, the vaccinated mice exhibited a modest yet consistent trend toward increased sensitivity to acetylcholine-mediated relaxation. While this phenomenon may not be clinically relevant, it warrants further investigation.

Overall, our study did not reveal any adverse effects of endurance exercise in the context of mRNA COVID-19 vaccination, despite its well-established immunomodulatory properties. On the contrary, both local and systemic responses tended to be lower in the vaccinated and exercising group. When considering exercise as an immunomodulator, our results add to previous work demonstrating the safety of SARS-CoV-2 vaccination in populations with different immune statuses (e.g., autoimmune diseases) and those undergoing immunomodulatory treatments [41,42].

## 5. Study Limitations

The clinical translatability of our results may be hindered by the vaccine’s administration route and disproportional dosing. Firstly, the BNT162b2 mRNA vaccine was administered intravenously to all the mice instead of via the intended intramuscular route, according to the vaccination protocol previously reported by Li et al. [9]. While intramuscular vaccination is known to elicit an adaptive immune response in the lymph nodes draining the injection side, the effects of the direct entrance and subsequent systemic distribution of lipid nanoparticle (LNP)-encapsulated mRNA on the immune response remain relatively unexplored [43,44]. Secondly, with each immunization, the mice received a disproportionately larger dose compared to what is typically administered to adolescent and adult humans (i.e., 30 µg BNT162b2 mRNA). Furthermore, as we utilized the BNT162b2 mRNA vaccine, our results cannot be generalized to other (mRNA) COVID-19 vaccines.

Our study exclusively used male mice, given previous research reporting a sex bias in myocarditis following mRNA COVID-19 vaccination, with a higher prevalence among males, particularly at a young age [24]. This sex and age bias also applies to viral myocarditis [45]. Furthermore, by excluding female mice, we aimed to eliminate potential influences of the estrous cycle on various aspects of our study, including exercise performance, the immune system, and the immune response to vaccination [46,47,48,49].

Although not yet fully characterized, LNP-mRNA formulations possess immunogenic properties [50]. The unvaccinated mice in our study received vehicle injections of normal saline, which, on the other hand, is not immunogenic. Ideally, these mice should have been administered an LNP-encapsulated non-coding RNA, using a formulation matched to the BNT162b2 mRNA vaccine. Moreover, increasing evidence suggests the role of the SARS-CoV-2 spike protein and its interaction with the angiotensin-converting enzyme (ACE2) in the pathogenesis of myocarditis post mRNA COVID-19 vaccination [51,52]. Since the spike protein binds less effectively to ACE2 in mice compared to humans, myocarditis in mice developed after mRNA COVID-19 vaccination might inherently not fully replicate the human disease pathology. In this context, human ACE2 knock-in and transgenic mice could offer a more dependable approach for investigating myopericardial injury induced by the novel mRNA COVID-19 vaccines [53,54,55].

## 6. Conclusions

Our data provide evidence for the safety of mRNA COVID-19 vaccination, particularly with regard to the adverse event of acute myopericarditis. Despite employing a cardiac injury-sensitive mouse strain, an intravenous administration route, and disproportionately high dosing, no evidence of myopericarditis was observed. Although myocardial inflammatory foci were present, these are suspected to be an inherent trait of the mouse strain used, and unrelated to vaccination or exercise status. Our study does not establish the contributing role of physical exercise to the development of pro-inflammatory cardiac changes post mRNA COVID-19 vaccination.

## Figures and Tables

**Figure 1 vaccines-12-00966-f001:**
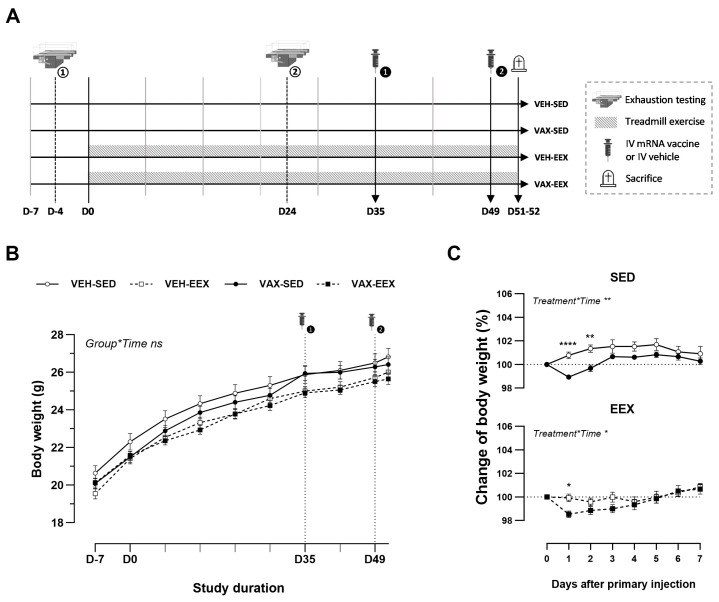
Experimental design and body weight changes. (**A**) Schematic overview of this study’s design. (**B**) Evolution of body weight (g) during the course of observation. A linear mixed model (LMM) was constructed with body weight as the dependent variable, and the group and time as the categorical fixed effects. The interaction term group*time was added, revealing that the change in body weight over time did not differ between the groups (*p* = 0.994). (**C**) Change in body weight (%) after the first injection with BNT162b2 mRNA or vehicle. Vaccination induced a mild, transient drop in body weight in both the sedentary and exercising mice. A repeated measures two-way ANOVA with Šidák’s multiple comparisons test was performed. The data (**B**,**C**) are represented as the mean ± SEM. Group sizes: VEH-SED: *n* = 17; VEH-EEX: *n* = 18; VAX-SED: *n* = 27; VAX-EEX: *n* = 26. EEX: exercise; SED: sedentary. * *p* < 0.05, ** *p* < 0.01, and **** *p* < 0.0001, ns = non significant.

**Figure 2 vaccines-12-00966-f002:**
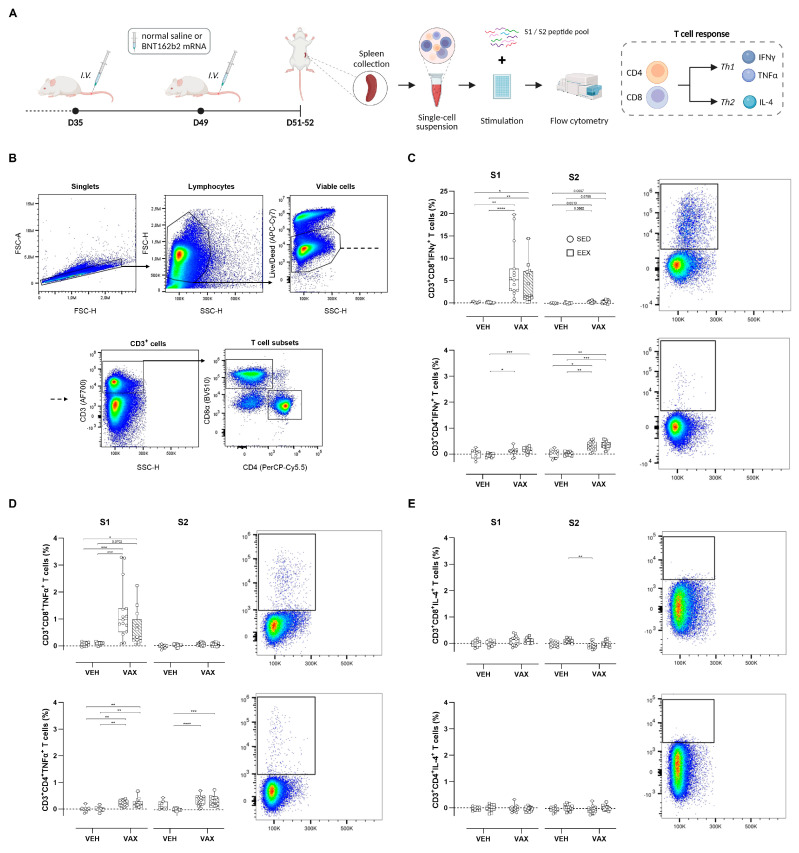
Antigen-specific T cell responses at euthanasia. (**A**) At euthanasia, mouse splenocytes were isolated from both vaccinated and unvaccinated mice and ex vivo stimulated with S1 or S2 peptide pools, followed by flow cytometric analysis. (**B**) General gating strategy for T cell subset (CD4^+^ and CD8^+^) identification. (**C**–**E**) Percentages of CD4^+^ and CD8^+^ cells expressing IFNy, TNF-α, or IL-4. Representative plots of each individual cytokine-specific T cell population are illustrated. For each individual animal, percentages per cytokine represent response measured after peptide stimulation, subtracted by unstimulated control (DMSO only). Kruskal–Wallis *H* test with Dunn’s multiple comparisons test. Group sizes: VEH-SED: *n* = 8; VEH-EEX: *n* = 10; VAX-SED: *n* = 15; VAX-EEX: *n* = 13. Data are represented as boxplots. IFNγ: interferon gamma; IL-4: interleukin-4; TNFα: tumor necrosis factor alpha. * *p* < 0.05, ** *p* < 0.01, *** *p* < 0.001, and **** *p* < 0.0001.

**Figure 3 vaccines-12-00966-f003:**
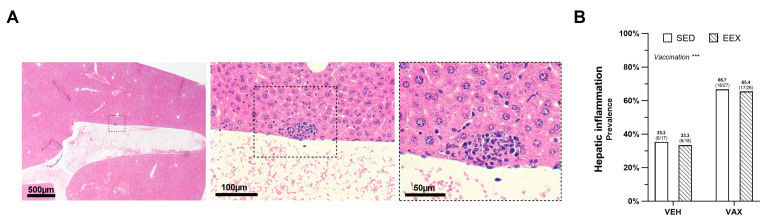
Hepatic inflammation at euthanasia. (**A**) Typical morphology of inflammatory lesion in liver at euthanasia on hematoxylin and eosin (HE) staining. (**B**) Frequency of hepatic inflammation at euthanasia as determined by histopathological scoring of HE-stained liver sections. Logistic regression model was constructed, which included exercise, vaccination, and their interaction as independent variables. Hepatic inflammation, defined as the presence of minimal portal and lobular inflammation, was used as dependent variable. Model indicated significant effect of vaccination on hepatic inflammation (*p* = 0.004) accounting for exercise, while no effect of exercise (*p* = 0.878), nor of their interaction was observed (*p* = 0.974). Data are represented as frequency histograms. Group sizes: VEH-SED: *n* = 17; VEH-EEX: *n* = 18; VAX-SED: *n* = 27; VAX-EEX: *n* = 26. *** *p* < 0.001.

**Figure 4 vaccines-12-00966-f004:**
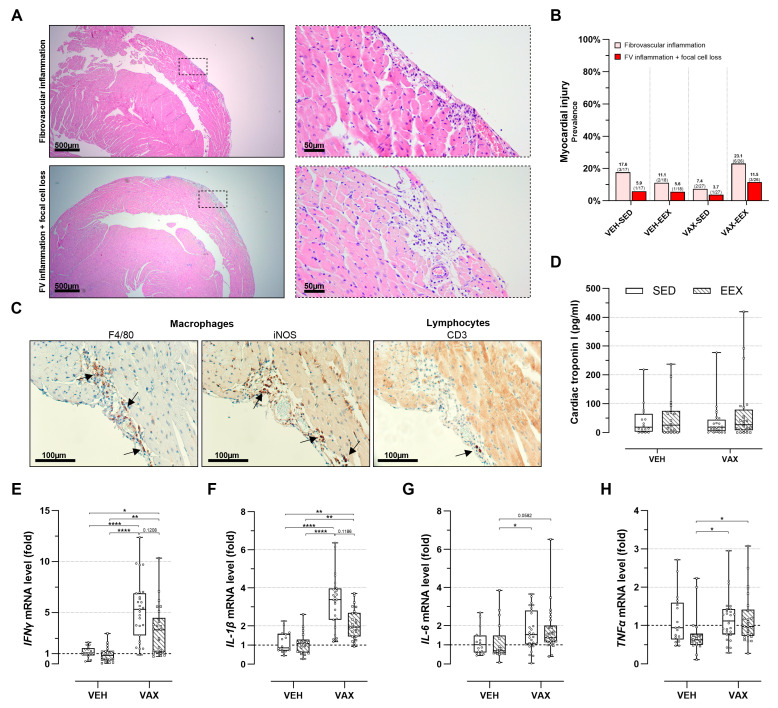
Cardiac injury at euthanasia. (**A**) Representative images of inflammatory heart lesions at euthanasia on hematoxylin and eosin (HE) staining, revealing fibrovascular inflammation alone (upper panels) or accompanied by focal cell loss (lower panels). (**B**) Frequency of cardiac injury as determined by histopathological scoring of HE-stained heart sections. Logistic regression model was constructed, including exercise, vaccination, and their interaction as independent variables. Cardiac injury, defined as presence of fibrovascular inflammation, either in conjunction with focal cell loss or not, was used as dependent variable. Likelihood ratio test was performed comparing this model to null model, with all mice having same odds of cardiac injury, and revealed no significance (*p* = 0.198). Group sizes: VEH-SED: *n* = 17; VEH-EEX: *n* = 18; VAX-SED: *n* = 27; VAX-EEX: *n* = 26. (**C**) Immunohistochemical characterization of observed inflammatory heart lesions at euthanasia. Infiltrates were found to have similar density and composition, mainly consisting of macrophages (F4/80^+^) of pro-inflammatory subtype (iNOS^+^), with sporadic presence of lymphocytes (CD3^+^) The arrows indicate immunopositive stained areas. (**D**) Cardiac troponin I (cTnI) plasma levels as marker for cardiac injury. At euthanasia, no significant differences were observed between experimental groups (*p* = 0.871). Group sizes: VEH-SED: *n* = 16; VEH-EEX: *n* = 18; VAX-SED: *n* = 27; VAX-EEX: *n* = 24. Kruskal–Wallis *H* test. (**E**–**H**) Pro-inflammatory cytokine expression levels in hearts at euthanasia as determined by RT-qPCR. mRNA expression levels are expressed as fold change over sedentary vehicle group (VEH-SED). Results are normalized against GAPDH and β-actin. Kruskal–Wallis *H* testing revealed statistically significant differences between all groups (IFNγ (*p* < 0.0001), IL-1β (*p* < 0.0001), IL-6 (*p* = 0.0118), and TNFα (*p* = 0.0150)). Pairwise comparisons between all groups were performed by Dunn’s multiple comparisons test. Group sizes: VEH-SED: *n* = 17; VEH-EEX: *n* = 18; VAX-SED: *n* = 27; VAX-EEX: *n* = 26. Data are represented as (**B**) frequency histograms or as (**D**,**E**) boxplots. Individual values are additionally shown. FV: fibrovascular; IFNγ: interferon gamma; IL-1β: interleukin-1 beta; IL-6: interleukin; iNOS: inducible nitric oxide synthase; TNFα: tumor necrosis factor alpha. * *p* < 0.05, ** *p* < 0.01, and **** *p* < 0.0001.

**Figure 5 vaccines-12-00966-f005:**
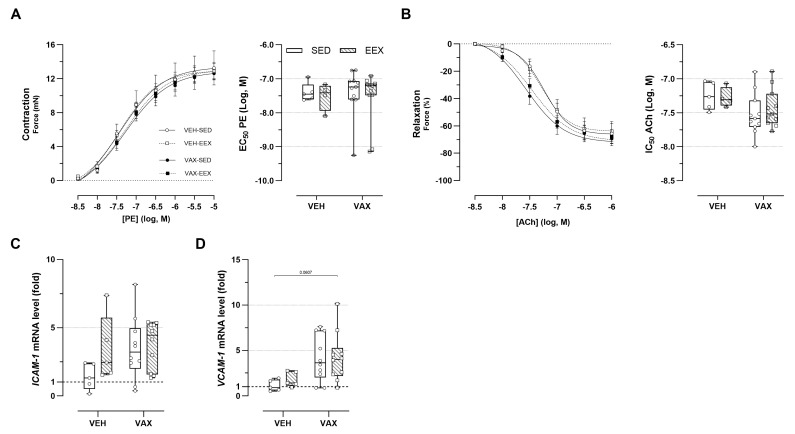
Vascular function and inflammation at euthanasia. (**A**,**B**) Concentration–response curves and half-maximal effective/inhibitory concentrations for (**A**) phenylephrine (PE)-induced vasocontraction and (**B**) acetylcholine (ACh)-induced endothelium-dependent vasorelaxation. At none of the concentrations were group differences in relaxation or contractility observed, nor were significant differences observed in sensitivity of vascular smooth cells for PE or ACh. Group sizes: VEH-SED: *n* = 5; VEH-EEX: *n* = 4; VAX-SED: *n* = 11; VAX-EEX: *n* = 11. Kruskal–Wallis *H* test with Dunn’s multiple comparisons test. (**C**,**D**) Adhesion molecule expression levels in aorta at euthanasia as determined by RT-qPCR. mRNA expression levels are expressed as fold changes over sedentary vehicle group (VEH-SED). Kruskal–Wallis H testing revealed statistically significant differences between all groups for VCAM-1 (*p* = 0.0297), but not ICAM-1 (*p* = 0.0913). Pairwise comparisons between all groups were performed by Dunn’s multiple comparisons test. Group sizes: VEH-SED: *n* = 5; VEH-EEX: *n* = 5; VAX-SED: *n* = 10; VAX-EEX: *n* = 10. Data are expressed as concentration–response curves or boxplots. Individual values are additionally shown. EC_50_: half-maximal effective concentration; ICAM-1: intercellular adhesion molecule 1; IC_50_: half-maximal inhibitory concentration; VCAM-1: vascular cell adhesion molecule 1.

## Data Availability

The data underlying this article will be shared upon reasonable request to the corresponding author.

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
