# Peer review of "Endurance Exercise Does Not Exacerbate Cardiac Inflammation in BALB/c Mice Following mRNA COVID-19 Vaccination"

_vaccines, 2024, doi:10.3390/vaccines12090966_

Round 1
Reviewer 1 Report
Comments and Suggestions for Authors
This preclinical study is particularly noteworthy in light of the intense debate surrounding vaccines such as the BNT162b2 mRNA vaccine and the risk of myopericarditis.
The number of experiments conducted is significant, and the methodology is robust. The figures are highly illustrative and well-executed. The cardiac evaluation has been thorough and detailed.
Some minor considerations:
- I would appreciate an exposition in the introduction on how the BNT162b2 mRNA vaccine has also been evaluated using clustering analysis in populations immunomodulated by drugs (it has often been mentioned that physical activity is immunomodulatory) while maintaining a certain level of safety. I recommend a study on a group of pathologies under extensive biological treatment, which has been the subject of much debate. An interesting study to discuss, which I suggest to the authors, is one conducted with the same BNT162b2 mRNA vaccine (https://pubmed.ncbi.nlm.nih.gov/36047032/);
- The paragraph on the analyses is probably somewhat sparse; I would enrich it.
Reviewer 2 Report
Comments and Suggestions for Authors
The current study titled “Endurance exercise does not exacerbate cardiac inflammation in BALB/c mice following mRNA COVID-19 vaccination” Ref: 3134219, deals with an important subject. Although previous reports discussed this subject mentioning the cardiac inflammation, the present article considered a well-organized methodology for young male mice administrating the vaccinations (two doses of BNT162b2 mRNA) intravenously not intramuscular as usually considered to humans. Minor revisions are needed.
- Although the study is well presented, poor/short discussions for the observations were noticed.
- Body weigh decrease of animals after vaccination should be discussed/explained.
- Animal death (five mice, VEH-SED: n=2, VEH-EEX: n=1, VAX-SED: n=2) after receiving vaccine seems a serious problem for this study.
- The side effects discovered (e,g. hepatic inflammation), that seems a serious issue associated with vaccination, should be discussed and justified.
